# The Role of the Federal Reserve in the U.S. Housing Crisis: A VAR Analysis with Endogenous Structural Breaks

**Mahua Barari** [1,*] **and Srikanta Kundu** [2]

1    Department of Economics, Missouri State University, Springfield, MO 65897, USA
2    Centre for Development Studies, Prasanth Nagar, Ulloor, Medical College P.O., Trivandrum, Kerala 695011, India
*    Correspondence: mahuabarari@missouristate.edu

**Abstract:** This paper reexamines the role of the Federal Reserve in triggering the recent housing crisis. Specifically, we explore if the relationship between the federal funds rate and the housing variables underwent structural changes in the wake of the housing crisis. Using quarterly data spanning 1960–2017, we estimate a VAR model involving federal funds rate, real GDP growth and a housing variable (captured by house price inflation or residential investment share or housing starts) and conduct time series analysis for the pre- and post-crisis periods. While previous studies mostly set break-dates based on events known a priori to split the full sample to subsamples, we endogenously determine structural break points occurring at multiple unknown dates. Our Granger causality analysis indicates that the federal funds rate did not cause house price inflation, although it caused residential investment share and housing starts in the pre-crisis period. In the post-crisis period, the real GDP growth caused residential investment and housing starts while house price inflation had a momentum of its own. Our impulse response and forecast error variance decomposition analysis reinforce these results. Overall, our findings suggest that housing volume fluctuates more than house prices over the business cycle.

**Keywords:** housing market; bubbles and crashes; time series econometrics; structural breaks; monetary policy

---

## 1. Introduction

There is a growing body of literature on the possible causes of the housing crisis of the past decade that eventually morphed into a full-blown recession during 2007–2009. The recession was of a magnitude not witnessed since the Great Depression of the 1930s and hence came to be known as the Great Recession. Naturally, there was a surge in interest in exploring the origin of this crisis to prevent its recurrence in future.

Mostly, four possible explanations have been offered for the housing bubble which burst in 2006 (See McDonald and Stokes 2013c, 2015). First, there are those who blamed the financial innovations of the 1990s and early 2000s giving rise to lax credit standards and greater access to mortgage credit for high-risk subprime borrowers. Second, some researchers pointed to the growing US current account deficit requiring a massive capital inflow from abroad. The 'global savings glut'[1] of the 1990s helped

---

1    Bernanke (2005) put out the hypothesis that during the 1990s, a combination of diverse forces created a significant increase in the global supply of saving (emanating in large part from China and other Asian emerging market economies), which helped

in channeling funds to the US and keeping the long-term interest rate under check. The third group placed the blame on a classic asset price bubble fueled by expectations of higher and higher home prices in future. Finally, there are those who held the Federal Reserve responsible for the run up in house prices by setting the target for the federal funds rate at very low levels during 2001–2004.

Our study is based on the explanation that focuses on the loose monetary policy of the Federal Reserve. We examine whether the interest rate policy of Federal Reserve played any role in the formation of the housing bubble and if so, how did it change since the bubble burst in 2006. Since structural stability in the relationship among variables is critical for such policy evaluation and analysis, we specifically investigate if the relationship between the federal funds rate, real GDP and the housing variables has been stable for all time periods or if there have been structural breaks in the relationship over time.

Controversy surrounds the role of the Federal Reserve to this day. Several papers have been written ranging from holding Federal Reserve primarily responsible to somewhat responsible to not responsible at all. McDonald and Stokes (2013b) provide an extensive literature review for the interested reader. For example, Taylor (2007) led the group that held monetary policy of Federal Reserve largely responsible for the turmoil in the housing market. Subsequently Leamer (2007), McDonald and Stokes (2013a, 2013b, 2015), Iqbal and Vitner (2013) arrived at a similar conclusion. On the other side of the aisle, Bernanke (2010) and Dokko et al. (2011) among others did not find evidence of any role played by the Federal Reserve in triggering the crisis. Bernanke (2010), in particular, considered low federal funds rate of 2002–2004 to be an appropriate response in the backdrop of a jobless recovery from the recession of the early 2000s as well as a growing concern of Japan style deflation taking hold. Shiller (2008) held a less extreme position arguing that although the period of very low funds rate coincided with rapidly rising house prices, the monetary policy was not necessarily an exogenous cause of the bubble. Still he agreed that the rate cuts might have boosted the housing boom beyond what would have otherwise occurred. Miles (2014) also held the Federal Reserve somewhat responsible while also attributing part of the blame to the global savings glut of the 1990s that was beyond the control of the Federal Reserve.

Some of these studies did focus on how the relationship between monetary policy and housing market variables has evolved over time. Iqbal and Vitner (2013) considered structural change caused by IT revolution and its subsequent bust in 2000–2001 to be a major contributor to the housing bubble. They used a dummy variable to isolate the time-period driving structural changes based on prior knowledge of historical events. McDonald and Stokes (2013a), on the other hand, carried out Granger causality tests between federal funds rate and house prices over different sub samples by choosing break dates based on known historical facts and concluded that there was a change in the relationship between funds rate and house prices after 2000. However, as Miles (2014) pointed out, "— changing the sample in this manner could lead to accusations, of 'cherry picking' different sample dates based on knowledge of the data and eventually would lead to inference problems". To avoid data mining, Miles (2014) performed the Andrews-Quandt endogenous structural break test (Andrews 1993) on the models. He found evidence of structural breaks in the impact of federal funds rate and the 30-year mortgage rate on both the house prices and residential fixed investment during the 1980s and the first half of the past decade.

While applying a test for structural break that endogenously determines the break point instead of guessing it based on prior knowledge is definitely a step in the right direction, Andrews and Quandt break test has its own share of limitations. First of all, it is a test for a single structural break point only and second of all, this test can only be applied to a single equation model. When all variables in the model are endogenous and may be subject to multiple structural breaks, Andrews-Quandt test will be inadequate. Qu and Perron (2007) method, by contrast, is much broader in scope as it allows for multiple structural changes occurring at unknown break dates within a system of equations.

---

explain the persistently low longer-term interest rates in the mid-2000's while the Federal Reserve was raising short-term interest rates.

We add a new dimension to the literature by first applying the Qu and Perron structural break test to endogenously determine break dates before splitting the full sample into pre- and post-crisis period to analyze the role of Federal Reserve in the housing crisis.

Following McDonald and Stokes (2013a, 2013b, 2013c) and Dokko et al. (2011), we apply a vector autoregression (VAR) model since the variables involved are endogenous in nature. We estimate a small dimensional VAR model consisting of federal funds rate, real GDP and a housing market specific variable using quarterly data spanning 1960:Q1 to 2017:Q4 where Q1, Q2, Q3 and Q4 refer to quarter 1, quarter 2, quarter 3 and quarter 4 respectively from here on. We select three alternative measures of housing market activity to distinguish between price and volume of construction. Model 1 uses nominal house prices while Model 2 focuses on the share of residential investment in GDP. Finally, Model 3 uses housing starts as an indicator of housing market activity. First, we apply the Qu and Perron (2007) break test to examine if structural changes have occurred in all three models and if so, at what dates. Second, we split the sample into segments using the break dates identified by Qu and Perron (2007) tests for each model, specifically focusing on the pre and post housing crisis period. This, in turn, ensures stability in relationship within each subsample which is critical for policy evaluation and analysis. Finally, we conduct a variety of multivariate time series analysis for the pre- and post-crisis period (including multivariate Granger causality tests, impulse response analysis and forecast error variance decomposition) to make inference about role of Federal Reserve in the crisis. All three models exhibit structural breaks in the relationship between variables. We find evidence of one statistically significant break in Model 1, four statistically significant breaks in Model 2 and as many as five statistically significant breaks in Model 3. It is worth noting that the 'potential' break dates obtained using Qu and Perron (2007) method are similar across models although not all of them are statistically significant. The break date that is significant across all three models occurs at 1986:Q4 (or 1987:Q1); a period often used in the literature as a probable break point because of the financial deregulation of the early 1980s. Models 2 and 3 each exhibit three additional break dates at 1969:Q1, 2000:Q1 (or 1999:Q4) and 2008:Q4 respectively. The break date in the late 1960s could be attributed to the establishment of Government Sponsored Enterprises (GSE) such as Fannie Mae and Freddie Mac[2] around that time which made mortgage financing more affordable and thus enhanced liquidity in this market. The break date of 2000:Q1 coincided with the tech boom and bust in the late 1990s and early 2000s while the 2008:Q4 coincided with the Great Recession following the collapse of the housing market. Finally, we find a fifth break point in Model 3 at 1977:Q4 which overlaps with the great inflation of the 1970s.

We use these endogenously determined break dates to split the full sample into subsamples, focusing on the last two subsamples. Based on the potential break dates from all three models, we choose 2000Q1–2008Q4 to be the pre-crisis period and 2009:Q1–2017:Q4 to be the post-crisis period. Our key findings can be summarized as follows. We do not find evidence of federal funds rate affecting house price inflation, either in the pre or post-crisis period. Our results contradict McDonald and Stokes (2013a, 2013b, 2013c) findings in this regard who have held the loose interest rate policy of the Federal Reserve responsible for the run-up in house prices. However, we find evidence of Federal Reserve influencing residential investment share and to some extent, housing starts in the pre-crisis period, both measuring of volume of construction activity in the housing market. In the post-crisis period, we observe a diminished role of the Federal Reserve in influencing the housing market variables. Instead, the macroeconomy as captured by real GDP growth gains strength in influencing housing variables although its impact is confined to residential investment share and housing starts again. House price inflation is mostly explained by its own past shocks in the post-crisis period suggesting a

---

2    Federal National Mortgage Association is commonly known as Fannie Mae while the Federal Home Loan Mortgage Corporation is known as the Freddie Mac.

built-in momentum. Overall, our findings support Leamer's assertion that housing sector experiences more of a volume cycle than a price cycle.

Our paper is organized as follows. In Section 2, we summarize previous literature and place our work in context. In Section 3, we provide a brief theoretical description of Qu and Perron (2007) method. In Section 4, we start with a discussion of our VAR models and data. This is followed by a detailed discussion of our empirical results. We offer our concluding remarks in Section 5.

## 2. Previous Literature

There is a growing body of literature assessing the role of the Federal Reserve in triggering and deepening the housing crisis of the past decade. As Dokko et al. (2011) pointed out, the residential investment share in GDP surged to 6.25% by late 2005, the highest share in a half century. This substantial increase in housing market activity in turn created a huge run-up in house prices. However, the bubble burst in 2006 when both residential investment share and house prices collapsed. During 2000–2006, the Federal Reserve first lowered the target for the federal funds rate from 6.5% in December 2000 down to 1% in June 2003. Subsequently it increased the target rate steadily until it reached 5.25% in June 2006. Obviously, question arose as to whether the interest rate policy of the Federal Reserve was responsible for the huge increase and subsequent decrease in housing market activity.

Taylor (2007) led the group that held the Federal Reserve primarily responsible for the crisis. According to Taylor, the low interest rates made the housing finance cheap and attractive and led to a surge in housing starts which reached a 25-year high by the end of 2003 and remained high until the downturn began in early 2006. Taylor (2007) estimated a simple housing starts equation from 1959 to 2007 with the federal funds rate as the explanatory variable and found a statistically significant effect of the lagged value of the funds rate on housing starts. Next, he used a counterfactual scenario to address what would have happened had the Federal Reserve followed the interest rate prescribed by Taylor Rule as given below.

$$i = p + 0.5y + 0.5(p - 2) + 2$$

where $i$ is the federal fund rate, $p$ is the rate of inflation over the previous four quarters, $y$ is the percent deviation of real GDP from the target, i.e., $y = 100(Y - Y^*)/Y^*$ where $Y$ is real GDP and $Y^*$ is trend of real GDP [see, Taylor (1993) for details]. During 2002–2006, the Taylor rule based interest rate was calculated to be much higher than the actual funds rate. Based on counterfactual simulation results, Taylor concluded that the deviation of the actual funds rate from the Taylor rule might have been the cause of the boom and bust in housing starts and house price inflation.

McDonald and Stokes (2013a, 2013b, 2013c, 2015) arrived at similar conclusions using a variety of model specifications in a VAR framework. McDonald and Stokes (2013a) applied the Granger causality test to study the effect of the federal funds rate on the Case Shiller Index of house prices in a two-variable VAR model and concluded that the Federal Reserve's low interest rate during 2001–2004 helped cause the housing bubble. McDonald and Stokes (2013b) expanded the original VAR framework to also include the foreclosures, the unemployment rate and the 30-year standard mortgage interest rate and demonstrated using impulse response functions that negative shocks to the funds rate increased house prices. Closely following on the prior two papers, McDonald and Stokes (2013c) furthered the analysis by studying the effect of 30-year mortgage rate on house prices and concluded that unlike shocks to the funds rate, shocks to the mortgage rate did not have any statistically significant impact on house prices. McDonald and Stokes (2015) expanded the VAR framework to include federal deficit as a proxy for fiscal policy as well as the interest rate on adjustable rate mortgages and a measure of net capital inflow. The most important finding of this study was that the house prices were impacted by the federal funds rate and the interest on adjustable rate mortgage but not by the interest rate on 30-year mortgage.

Leamer (2007, 2015) held the view that the housing downturn was built by the Federal Reserve. He put forward the explanation that the Federal Reserve can mostly affect the timing of home building but not the total over time. The low interest rate of the Federal Reserve can transfer home construction

forward in time when construction that did not take place during recession occurs in its aftermath. On the other hand, low interest rate can also transfer construction backward in time by shifting construction that might have occurred at some point in future backward in time into the current period. In Leamer's opinion, the low federal funds rate of 2002–2004 did not transfer construction forward in time since there was nothing to transfer. Instead, it had the effect of transferring construction backward in time by building more in 2003–2005 at the expense of building less in 2006–2008 thereby sowing the seed for the crisis.

Furthermore, Leamer observed that housing is the single most critical part of the U.S. business cycle. In fact, nine of the past eleven recessions were preceded by a housing downturn. This is because according to Leamer, homes prices primarily experience a volume cycle, not a price cycle. In the face of weak demand, home prices are slow to drop, instead volume decreases more. For real GDP growth and the associated job creation, what matters is the volume of activity in the housing market. Leamer acknowledged that the most recent housing crisis witnessed a sharp drop in house prices. This was due to loose lending standards that allowed the homeowners under financial distress to turn over ownership of property to the lending agencies that are not reluctant sellers. Still Leamer (2015) concluded that "I stick with my view that housing experiences a volume cycle not a price cycle, and that 2007–09 was a never-to-be-repeated exception".

On the other side of the isle, Dokko et al. (2011) were of the view that the Federal Reserve's interest rate policy was well aligned with the goals of the policy makers and not the primary contributor to the extraordinary surge in housing market activity. They employed two empirical models to support their view—The Federal Reserve Board's FRB/US model and a reduced form seven variable VAR. Their simulation results based on the FRB/US model suggested that a much tighter monetary policy than what was pursued would have resulted in an unemployment rate far higher than what was realized at the time. For the VAR framework, Dokko et al. (2011) used a conditional forecast approach, conditional on other variables that entered the VAR. Their simulation exercises suggested that monetary policy could not have played a major role in the housing market boom.

Bernanke (2010, 2015) defended the accommodative policy stance of the Federal Reserve during 2002–2004 by saying it was what was needed at the time to address the jobless recovery from the recession that technically ended in 2001. Furthermore, Federal Reserve's policy response also partly reflected its concerns about a Japan type deflation setting in. In response to Taylor's criticism that the funds rate was much lower than that prescribed by the Taylor's rule, Bernanke questioned the methodology for computing the output or inflation gap in the Taylor rule. The deviation of the actual funds rate from the Taylor prescribed rate was much smaller when the forecast values of inflation or output were considered instead of their currently observed values. Using International Monetary Fund (IMF) data from a sample of 20 industrial countries, Bernanke also demonstrated that the relationship between monetary policy and house price appreciation was quite weak across countries. Bernanke concluded that the magnitude of house price increases during the early half of the past decade was too large to be explainable by accommodative monetary policy alone. What was needed instead was regulatory reform to control the explosion of exotic types of mortgages and the associated lax lending standards.

Shiller (2008) and Miles (2014) took a stand in the middle. Shiller alluded to the fact that the house prices started appreciating in the late 1990s, much before the period of low interest rate of 2002–2004. In fact, the housing boom was already accelerating in the late 1990s amidst tightening of the interest rate by the Federal Reserve. The rise of alternative mortgage products and loose lending standards were more of a response to rising prices and not the other way around. In that sense, it was a speculative boom that was fueled by expectations of ever-rising house prices. For a while, rising house prices became a self-fulfilling prophecy; eventually further appreciations could not be sustained, and house prices collapsed. That said, rate cuts by the Federal Reserve during 2002–2004 certainly deepened the crisis albeit it was not the exogenous cause of the crisis.

Miles (2014) emphasized the need to incorporate a proxy for long-term interest rate in addition to the federal funds rate. He alluded to the fact that long-term interest rates were held down even when the Federal Reserve was raising rates during 2004–2006 due to global factors beyond the control of the Federal Reserve. Using a reduced form model where the cyclical components of the housing variables were regressed on the lagged cyclical components of funds rate and a standard 30-year mortgage rate, Miles found the long-term rate to have independent and sometimes greater predictive power for housing than the funds rate. He concluded that while his results did not necessarily exonerate the Federal Reserve, they did point to other global factors beyond Federal Reserve's control that kept the long-term interest at low levels for an extended period thereby sustaining the crisis.

The objective of our study is to reassess the role of the Federal Reserve in the recent housing crisis with special focus on structural stability in the relationships among variables. While some of the earlier papers discussed the possibility of structural changes occurring in relationships between these variables, except for Miles (2014), none of these papers applied statistical tests to endogenously determine break dates. For example, McDonald and Stokes (2013a) performed Granger causality tests over different subsamples (1987–2000 and 2000–2010) by arbitrarily choosing break points based on historical facts and concluded that the impact of funds rate on house prices strengthened over time. Iqbal and Vitner (2013) tested the structural change hypothesis by assigning a dummy variable equal to one for the period 2001:Q4–2006:Q3 and 0 otherwise. They found a statistically significant effect of the dummy variable on housing starts and house prices. Miles (2014), on the contrary, pointed out that choosing break points based on known facts may give rise to misleading inference from hypothesis testing. Instead, he formally tested for parameter instability over time by applying the Andrew-Quandt test (Andrews 1993) for structural change and found evidence of significant structural breaks.

We concur with Miles that the break points in the relationships need to be determined endogenously within the model. This is because testing for parameter instability by splitting samples based on events known a priori is a form of data mining as the selection of the break date has been made after the data has been informally examined [See Hansen (1992)]. However, the Andrew-Quandt test is limited in scope as it can determine only one unknown break date in a single equation model. For our VAR model, we apply a more advanced test developed by Qu and Perron (2007) that can determine multiple unknown break dates in a multi equation framework. It is worth noting that Ahamada and Sanchez (2013) applied the Qu and Perron (2007) test to study the U.S. house price-macro link over time using a VAR framework. Using quarterly data from 1960:Q1 to 2009:Q3, they found evidence of two break points in the house price-real GDP relationship. However, they did not address the effect of the Federal Reserve's interest rate policy on the housing variables in their study.

In this backdrop, first we test for stability in relationships among variables against the alternative of multiple structural breaks in a VAR framework using Qu and Perron (2007) method. Second, if the null hypothesis is rejected, we identify the number of breaks and estimate the break dates. We split the full sample into pre- and post-crisis period using the estimated break dates thereby ensuring structural stability within subsamples. Finally, we analyze the impact of Federal Reserve's interest rate policy on housing variables by carrying out multivariate Granger causality tests for the pre- and the post-crisis period along with impulse response and forecast error variance decomposition analysis.

## 3. Qu-Perron Method for Detecting Structural Breaks

Structural changes have always been an important concern in econometric modelling of time series data typically spanning several decades. As Bai (2000) pointed out, correctly detecting and identifying structural changes in the relationships is critical as it can have profound implications for policy evaluation and analysis. While earlier econometric studies on this topic included Chow (1960) and Quandt (1960), recent studies included Andrews (1993), Bai (1997), Liu et al. (1997), Bai and Perron (1998), Bai and Perron (2003) among others. The test devised by Andrews (1993) was the first test for structural break where the break date was considered unknown when testing the null hypothesis of no break against the alternative of a single break. Bai (1997) studied the least squares estimation of a

single change point in multiple regression and developed the asymptotic theory for it. Liu et al. (1997) studied multiple structural shifts in a linear regression model focusing on rate of convergence of the estimated break dates as well as the consistency of the selection criterion to determine the number of breaks. Bai and Perron (1998) extended previous work of Bai (1997) to include multiple structural changes occurring at unknown dates in a multiple regression model under much less restrictive set of assumptions than Liu et al. (1997). They studied theoretical issues in this paper pertaining to the limiting distribution of estimators and test statistics. Subsequently, in a companion paper, Bai and Perron (2003) considered practical issues related to empirical applications of the procedures and provided relevant algorithms for it.

To this date, only a handful of studies have dealt with multiple structural changes at unknown dates in a system of multivariate equations. Bai (2000) analyzed multiple structural changes in a segmented stationary VAR model focusing on the asymptotic behavior of break-point estimators. Qu and Perron (2007) provided a more comprehensive treatment of issues pertaining to estimation, inference and computation in case of multiple structural changes that occur in a variety of multiple equation time series models such as VAR, seemingly unrelated regression (SUR) and certain linear panel data models. Qu and Perron (2007) provided algorithms for structural changes resulting from changes in regression coefficients and/or the covariance matrix of errors across regimes. Furthermore, Qu and Perron (2007) considered both pure structural changes occurring due to changes in all regression coefficients across regimes as well as partial structural changes where only a subset of regression coefficients is allowed to vary but not all. In this study, we apply the Qu and Perron (2007) method for pure structural changes to our three alternative VAR models. What follows is a brief theoretical discussion of the Qu and Perron (2007) method for testing and estimating unknown break dates.

Consider the model given by

$$y_t = \left(I \otimes z_t'\right) S \beta_j + u_t$$

where we have $n$ equations and $T$ observations excluding the initial conditions. We consider $m$ structural breaks in the system. The break dates are $(T_1, T_2, \ldots, T_m)$. We also denote $T_0 = 1$ and $T_{m+1} = T$. $y_t = (y_{1t}\, y_{2t} \ldots y_{nt})'$ is the vector of $n$ variables. $z_t = (z_{1t}, z_{2t}, \ldots, z_{qt})'$ is the vector of lagged values of $y_t$ where we consider $q$ number of lags. The disturbance term $u_t$ has a mean value of 0 and covariance matrix $\Sigma_j$ for $T_{j-1} + 1 \leq t \leq T_j$. $I$ is the identity matrix. The matrix $S$ is of order $nq \times p$ with full column rank. It is the selection matrix with elements 0 or 1 specifying which regressors appear in each regression. The set of parameters in regime $j$ consists of the $p$ vector $\beta_j$ and $\Sigma_j$.

For the purpose of estimation, we can rewrite the model using convenient notation

$$y_t = x_t' \beta_j + u_t$$

for $T_{j-1} + 1 \leq t \leq T_j$ $(j = 1, 2, \ldots, m)$.

The model is estimated by restricted Quasi Maximum Likelihood method assuming that errors are serially uncorrelated.

Conditional on the given subsample, Gaussian likelihood—function can be written as

$$L_T(T, \beta, \Sigma) = \prod_{j=1}^{m+1} \prod_{t=T_{j-1}+1}^{T_j} f(y_t | x_t; \beta_j \Sigma_j)$$

where

$$f\left(y_t | x_t; \beta_j \Sigma_j\right) = \frac{1}{\left(2\pi |\Sigma_j|\right)^{1/2}} \exp\left[-\frac{1}{2}\left(y_t - x_t'\beta_j\right)' \Sigma_j^{-1}\left(y_t - x_t'\beta_j\right)\right]$$

and the likelihood ratio is

$$LR_T = \frac{\prod_{j=1}^{m+1} \prod_{t=T_{j-1}+1}^{T_j} f(y_t|x_t; \beta_j \Sigma_j)}{\prod_{j=1}^{m+1} \prod_{t=T_{j-1}^0+1}^{T_j^0} f(y_t|x_t; \beta_j^0 \Sigma_j^0)}$$

The aim is to obtain values of $T_1, T_2, \ldots, T_m$, $\beta, \Sigma$ that maximize $LR_T$ subject to the restrictions $g(\beta, vec(\Sigma)) = 0$.

Denoting log likelihood ratio and the restricted log likelihood ratio by $lr_T(\cdot)$ and $rlr_T(\cdot)$ respectively, we can write the objective function as follows.

$$rlr_T(T, \beta, \Sigma) = lr_T(T, \beta, \Sigma) + \lambda' g(\beta, vec(\Sigma))$$

And the final estimates are $\left(\widetilde{T}, \widetilde{\beta}, \widetilde{\Sigma}\right) = \arg \max_{(T_1, \ldots, T_m; \beta, \Sigma)} rlr_T(T, \beta, \Sigma)$.

Under a set of eight assumptions that are stated in details in Qu and Perron (2007), they have obtained that for $j = 1, 2, \ldots, m$, $v_T^2\left(\hat{T}_j - T_j^0\right) = O_p(1)$, and for $j = 1, 2, \ldots, m+1$, $\sqrt{T}\left(\hat{\beta}_j - \beta_j^0\right) = O_p(1)$ and $\sqrt{T}\left(\hat{\Sigma}_j - \Sigma_j^0\right) = O_p(1)$, where $v_T$ is either a positive number independent of $T$ or a sequence of positive numbers that satisfy $v_T \to 0$ and $T^{1/2} v_T / (\log T)^2 \to \infty$. They have also shown that the limiting distributions of conditional mean and covariance matrix of the errors in each regime estimated based upon unknown break dates are the same as in the case with known break dates.

Qu and Perron (2007) have proposed a few test statistics for testing and subsequently identifying multiple break points in a system of $n$ equations. In these tests, all time points except for some at the beginning and some at the end of the time series are considered probable break dates. The proportion of time points to be left out so that the first and the last sub-periods can be statistically distinguished is called the trimming parameter, and its value is often taken to be 0.15. The tests proposed by Qu and Perron are the following.

1.  The *Sup LR* test considers a likelihood ratio test for the null hypothesis of no structural break versus an alternative hypothesis of structural break with a pre-specified number of breaks, say $m$.
2.  The *UDmax* test and the *WDmax* test consider an equal weighting scheme and unequal weighting scheme, respectively, where weights depend on the number of regressors and the significance level of the tests. For these two tests, the alternative hypothesis is any number of breaks with some specified maximum.
3.  The $seq(l+1|l)$ test is a sequential test for the null hypothesis of $l$ breaks versus the alternative of $(l+1)$ breaks.

It should be obvious that the size and power of these tests are important issues for the purpose of final inference. Like Bai and Perron (1998, 2003), Qu and Perron (2007) have suggested the following useful strategy. First, a researcher should use either or both of the double maximum tests i.e., the *UDmax* test and the *WDmax* test to see if at least one break is present. If these tests indicate the presence of at least one break, then $seq(l+1|l)$ sequential break test can be constructed using global minimizers to determine the number of structural breaks. In our study, we first apply the $WD_{max}$ test to see whether there exists at least one structural break. If the null hypothesis of $WD_{max}$ is rejected, we then consider $seq(l+1|l)$ to find out the actual number of breaks.

## 4. VAR Models, Data and Results

### 4.1. VAR Models

We apply Qu and Perron (2007) method for estimating break points to three alternative VAR models. These models present relationship between three endogenous variables—the federal funds rate as a proxy for Federal Reserve's monetary policy stance, real GDP as a proxy for general macroeconomic

condition and a housing variable to track housing market activity. We use three alternative housing market variables—nominal house prices in Model 1, residential investment share in GDP in Model 2 and housing starts in Model 3. Following upon Leamer (2007, 2015), one of our goals in this paper is to investigate if Federal Reserve's policy had a similar impact on volume vis-à-vis prices in the housing market in the pre or post-crisis period. The residential investment share and housing starts in Models 2 and 3 capture volume of constructions in housing market as opposed to their prices.[3]

Our choice of housing variables is rooted in past studies. For example, McDonald and Stokes (2013a, 2013b, 2013c, 2015) used a nominal house price index in all their papers as they explained that borrowers in the housing market took the loans in nominal terms and defaulted in nominal terms. In a similar vein, Miles (2014) also used a nominal home price index in his study to analyze the role of Federal Reserve. Regarding residential investment share, Dokko et al. (2011) used it as a housing variable in their VAR study. Finally, with respect to housing starts variable, Taylor (2007) used it in his counterfactual simulation study to analyze the impact of Federal Reserve's interest rate policy.

For VAR estimation, the variables will have to be stationary, i.e., I(0). The real GDP and house prices are found to be non-stationary in our models given by I(1) while the funds rate, the residential investment share and housing starts are found to be I(0). The results of ADF tests are given in Table A1 of the Appendix A. We take the first difference of the log of real GDP and log of nominal house prices to make them stationary. Thus, our VAR models examine relationship between the funds rate, the real GDP growth and either house price inflation, residential investment share or housing starts which can be mathematically expressed as below.

$$y_t = C + A_1 y_{t-1} + A_2 y_{t-2} + \ldots + A_p y_{t-p} + \varepsilon_t$$

where $y_t = (\text{FFR}_t \ \Delta\text{LRGDP}_t \ X_t)'$ a $3 \times 1$ vector of dependent variables, $C$ is $3 \times 1$ vector of constants, $A_i \ (i = 1, \ldots, p)$ is $3 \times 3$ matrix of slope coefficients, $\varepsilon_t$ is the error term with $E(\varepsilon_t) = 0$, and $Var(\varepsilon_t) = \Sigma_{3 \times 3}$, and $p$ is a positive integer representing the number of lags. FFR and $\Delta\text{LRGDP}$ represent the federal fund rate and the growth rate of real GDP respectively. $X_t$ represents either house price inflation ($\Delta\text{LNHPA}$) or residential investment share (RIS) or housing starts (HS).

### 4.2. Data

We use quarterly data for a period ranging from 1960:Q1 to 2017:Q4. The underlying data generating mechanism is likely to have undergone structural changes over such a long horizon.[4] We retrieve all data except for house prices from the FRED database (https://fred.stlouisfed.org/) made available by the Federal Reserve at St. Louis. The historical data on house prices is obtained by averaging monthly data available in Shiller (2015) and the website maintained by the author (http://www.econ.yale.edu/~{}shiller/data.htm). The funds rate is measured by the end of period quarterly data on effective funds rate. The real GDP is measured in 2009 dollars. The residential investment share is calculated as the ratio of nominal residential investment to nominal GDP where residential investment represents seasonally adjusted private residential fixed investment. The data on housing starts is based on the total number of new privately-owned housing units started. For Qu and Perron (2007) break test, we use Gauss code available on Perron's personal website (http://people.bu.edu/perron/code.html). The rest of the estimations are carried out on Eviews software.

---

[3] A similar VAR model was also used by Ahamada and Sanchez (2013) in studying house price—macro relationship. Miles (2014) and McDonald and Stokes (2013c) used a standard 30-year mortgage rate in addition to the funds rate but arrived at different conclusions regarding the impact of long-term interest rate on house prices. Since our objective is to examine the role of the Federal Reserve and not the merit of global savings glut hypothesis, we decided against including long-term interest rate in our model.

[4] Ahamada and Sanchez (2013) that applied Qu and Perron (2007) test to study house price-macro link also used 1960:Q1 as the starting point. However, their sample ended in 2009:Q3.

*4.3. Discussion of Results*

4.3.1. Estimation of Break Dates—Pure Structural Change with No Linear Constraints

First, we carry out Qu and Perron pure structural break tests for all three VAR models where all regression coefficients are allowed to change across regimes, but covariance matrix stays the same. Since we are focusing on structural break in the relationship between housing market variables and the funds rate, we are reporting the break in mean parameters only and not the break in variance which is due to unobserved factors. Following Qu and Perron (2007) method, we start out with the WDmax test which tests the null hypothesis of no structural break against the alternative of unknown number of breaks. The results are given in Table 1 below.

**Table 1.** $WD_{max}$ test for up to 5 breaks: VAR with no linear restriction.

| test | ΔLRGDP, ΔLNHPA, FFR | ΔLRGDP, RIS,FFR | ΔLRGDP, HS, FFR |
|------|---------------------|-----------------|-----------------|
| *WDmax* | 41.7334 | 61.0579 | 60.4645 |

Critical value at 5% is 31.46. FFR, ΔLRGDP, ΔLNHPA, RIS and HS denote the federal funds rate, real GDP growth, house price inflation, residential investment share in GDP and housing starts respectively.

As Table 1 indicates, the calculated WDmax test statistics based on our sample exceed the 5% critical value of 31.46 for all three models. Therefore, we reject the null and conclude that at least one structural break or more is present in all our models. Following Qu and Perron, next we proceed to the sequential break test to narrow down the number of breaks as well as estimate the break dates. Here the test statistic is given by $(seq(l+1|l))$ which tests for $l+1$ breaks given that there are $l$ number of breaks. The hypothesis testing is sequential here in the sense that we move on to the next higher number of breaks only if null is rejected. The sequential search process stops when we fail to reject the null hypothesis of $l+1$ breaks given $l$ breaks. Given our sample size, we choose the maximum number of breaks to be 5 and 0.15 as the trimming parameter. The results of sequential break tests with 5% critical value are given in Table 2 below.

**Table 2.** The $Seq(l+1|l)$ break test for up to 5 breaks with no linear restriction.

| | ΔLRGDP, ΔLNHPA, FFR | | ΔLRGDP, RIS, FFR | | ΔLRGDP, HS, FFR | |
|--------|----------------|---------------------|----------------|---------------------|----------------|---------------------|
| **Test** | **Test Statistic** | **Break Dates** | **Test Statistic** | **Break Dates** | **Test statistic** | **Break Dates** |
| Seq(1\|0) | 34.103 | 1986Q4 (1986Q2–1987Q3) | 42.133 | 1987Q1 (1986Q4–987Q2) | 36.601 | 1987Q1 (1986Q4–1987Q2) |
| Seq(2\|1) | 26.774 | 2008Q4 (2008Q3–2009Q1) | 49.786 | 2008Q4 (2007Q2–2009Q2) | 61.811 | 2008Q4 (2008Q2–2009Q2) |
| Seq(3\|2) | 47.466 | 1969Q1 (1968Q3–1969Q3) | 108.621 | 1969Q1 (1968Q3–1969Q3) | 61.811 | 1977Q4 (1977Q3–1978Q1) |
| Seq(4\|3) | 12.319 | 1978Q1 (1977Q3–1978Q3) | 108.621 | 2000Q1 (1999Q2–2000Q4) | 57.172 | 1969Q1 (1968Q3–1969Q3) |
| Seq(5\|4) | 37.769 | 2000Q1 (1997Q4–2001Q4) | 0.0000 | 1978Q1 (1977Q3–1978Q3) | 38.209 | 1999Q4 (1998Q4–2000Q4) |

Critical values at 5% levels are 29.9606, 33.4526, 34.8442, 35.7791 and 36.5315 respectively. The 95% confidence intervals of the break dates are given in parentheses. The sequences of break dates are determined by repeated estimates of structural breaks by considering a maximum of 1 to 5 breaks. FFR, ΔLRGDP, ΔLNHPA, RIS and HS denote the federal funds rate, real GDP growth, house price inflation, residential investment share in GDP and housing starts respectively.

Based on Table 2, we can make the following observations. Model 1 involving funds rate, real GDP growth and house price inflation exhibits one significant structural break at 1986:Q4. Model 2 involving funds rate, real GDP growth and residential investment share in GDP exhibits four significant structural breaks at 1969:Q1, 1987:Q1, 2000:Q1 and 2008:Q4 in chronological order. The value of test statistic for $Seq(5|4)$ is 0.0 in Model 2 which according to the Gauss code implies that given the location of the

breaks from the global optimization with 4 breaks, there is no more place to insert an additional break that satisfy the minimum length requirement. Finally, Model 3 involving funds rate, real GDP growth and housing starts exhibits five significant structural breaks at 1969:Q1, 1977:Q4, 1987:Q1, 1999:Q4 and 2008:Q4. Since Model 3 reached the maximum number of breaks, we also looked at the value of test statistic for *Seq*(6|5). It turned out to be 0.0 indicating (per Gauss code) that there was no more place to insert an additional break. Clearly, the significant break dates across all three models are mostly identical and at most one quarter apart. Furthermore, albeit Model 1 contains only one significant break, the potential break dates found using Qu and Perron (2007) method for this model strongly overlap with the other two models.

These break dates that are determined endogenously within our VAR models are also backed by historical events. It is worth noting that Ahamada and Sanchez (2013) which applied the Qu and Perron (2007) method to study changes in house price-macro link found two breaks in their VAR models (for the sample period 1960:Q1–2009:Q3) occurring in the late 1960s and the mid-1980s. They explained the rupture in the relationship in the late 1960s in terms of the privation of Fannie Mae in 1968 and the creation of the Freddie Mac in 1970 that had a profound impact on the mortgage market. As regards break point in the mid 1980s, such a date has been used extensively in the literature on housing market studies that relied on predetermined break dates[5] since this period, historically speaking, coincided with the start of the period of "Great Moderation"[6]. This period also came on the heels of the financial deregulation of the early 1980s that led to deep securitization of the mortgage market facilitated by the GSEs. The break date we find in late 1990s or early 2000s has a few different explanations. First, as Shiller (2008) pointed out, the rise in home values dated back to 1997 that coincided with the elimination of federal capital gains tax on owner occupied homes making it cheaper for prospective homebuyers to speculate in the housing market. This was followed by the tech boom and bust of the late 1990s and early 2000s as well as 9/11 attack leading up to a mild recession in 2001. Finally, this period also witnessed a rapid expansion of sub-prime loans accompanied by lax credit standards which set the stage for the housing crisis. Finally, the break date of 2008:Q4 coincided with the Great Recession of the past decade which was preceded by a total collapse of the housing market.[7]

### 4.3.2. Time Series Analysis for Subsamples—Pre and Post-Crisis

Next, we consider time series evidence for assessing Federal Reserve's role in the housing crisis. To that effect, we use the estimated break dates to divide our sample into segments with special focus on the last two subsamples to highlight the pre and post housing crisis period. Using common subsamples across models, we decide on 2000:Q1–2008:Q4 as our pre-crisis period and 2009:Q1–2017:Q4 as our post-crisis period. While these break dates are not statistically significant in Model 1, these are still found to be potential break dates using Qu and Perron (2007) method.[8]

First, we estimate the VAR models for these sub samples. Using the Schwarz Information criteria (SIC) for lag selection[9], we lag each of the three endogenous variables in the model by one quarter. Our estimated VAR models, given in Tables A2–A4 in the Appendix A, show that the lagged value of federal funds rate does not have a significant impact on house prices in the pre-crisis period. However, it has a significant negative impact on the residential investment share at 1% level and housing starts

---

[5]   See Miles (2009) for a thorough literature review of studies that investigated structural breaks in relationship between housing and the rest of the economy over time. His own VAR analysis indicated that residential investment has become more important for the macroeconomy since the financial deregulation of the early 1980s.

[6]   The period of Great Moderation refers to the rule based monetary era from the mid-1980s until the early mid-2000s which was accompanied by substantial reduction in the volatility of macro variables such as real GDP, inflation and interest rate.

[7]   We found a fifth break date at 1977:Q4 only in Model 3 which coincided with the great inflation of the 1970s.

[8]   Furthermore, we find 1999:Q4 and 2008:Q4 to be significant break dates in Model 1 when we allow the covariance matrix of errors, in addition to the condition mean, to change across regimes.

[9]   $SIC(p) = \ln|\Sigma(p)| + \frac{\ln T}{T}pn^2$ where, $\Sigma(p)$ is the residual variance of the $VAR(p)$ model, $n$ is the number of variables in the $VAR$ model and $T$ is the total number of observations. The order of VAR is chosen by minimizing $SIC$ with respect to $p$.

at 10% level in the pre-crisis period. By contrast, we do not find the funds rate to be significant in explaining any of the three housing market variables in the post-crisis period. Instead, the lagged real GDP growth has a significant positive impact on the residential investment share and housing starts in the post-crisis period at 1% level and 5% level respectively.

To check the reliability of our VAR models, we carry out cross-autocorrelation test of the error terms next. Given the sample size in each period, we perform cross autocorrelations considering a lag and lead up to four quarter. In other words, we consider error term of the funds rate at time t and other error terms of each model at time $t - i$ and $t + i$, where $i = 0, 1, 2, 3, 4$. The plots of cross autocorrelations along with their two standard error band for the three models are given in Figure A1–A3 in the Appendix A. It is evident from these figures that there is no significant cross autocorrelation for both lead and lag up to fourth quarter. Obviously, there are contemporaneous correlations, as seen in the figures for $i = 0$, which are incorporated in the estimation of VAR models. The statistical insignificance of all lead and lag cross autocorrelations implies that our models are correctly specified.

To understand the causal nature of the relationships, we apply the multivariate Granger causality test next. This test is an improvement over the bivariate Granger causality test as it includes all other explanatory variables in the model in both null and alternative hypothesis when testing whether a particular variable Granger causes another variable or not. [See Hamilton (1994) for details] The test results are given in Table 3 below.

**Table 3.** Granger causality/Block exogeneity test in pre and post-crisis period.

| | | 2000Q1–2008Q4 | | 2009Q1–2017Q4 | |
|---|---|---|---|---|---|
| | | $\chi^2$ | *p*-Value | $\chi^2$ | *p*-Value |
| Model 1 | *FFR* ⇏ ΔLNHPA | 0.5662 | 0.45 | 0.2619 | 0.61 |
| | ΔLRGDP ⇏ ΔLNHPA | 0.4174 | 0.52 | 0.0218 | 0.88 |
| Model 2 | *FFR* ⇏ *RIS* | 11.3943 | 0.00 | 0.8318 | 0.36 |
| | ΔLRGDP ⇏ *RIS* | 1.9996 | 0.16 | 7.6998 | 0.01 |
| Model 3 | *FFR* ⇏ *HS* | 2.6238 | 0.10 | 0.0084 | 0.93 |
| | ΔLRGDP ⇏ *HS* | 0.0722 | 0.79 | 3.8164 | 0.05 |

FFR, ΔLRGDP, ΔLNHPA, RIS and HS denote the federal funds rate, real GDP growth, house price inflation, residential investment share in GDP and housing starts respectively.

As Table 3 indicates, the federal funds rate causes residential investment share in Model 2 (at 1% level of significance) and housing starts in Model 3 (at 10% level of significance) only in the pre-crisis period. Even more important, the funds rate does not cause house price inflation, neither in pre- nor in post-crisis period. In this respect, our findings contradict McDonald and Stokes (2013a) where they found evidence of Federal Reserve's interest rate causing house prices during 2000–2010 period. This prompted them to conclude that the Federal Reserve helped cause the housing bubble by lowering the interest rate and holding it down during 2001–2004. Instead, we find evidence of the federal funds rate causing volume of activities in the housing market in the pre-crisis period. Our findings here lend credence to Leamer's assertion that housing primarily experiences a volume cycle and not a price cycle. While Federal Reserve played a role in the housing crisis, it was limited to volume of construction and housing starts. In the post-crisis period, we do not find any evidence of Federal Reserve's interest rate policy causing house market activity. Instead, it is the real GDP growth that specifically causes volume indicators in the housing market in the post-crisis period.

We supplement the Granger causality tests with the impulse response functions next. This is because Granger causality tests can only comment on the presence or absence of causality but cannot determine the sign of the effect or the timing of it. The impulse response functions as derived from the VAR models trace the responsiveness of each series to its own shocks as well as shocks from other series in the model. Since the estimated impulse response functions can be sensitive to the ordering of

variables if there are contemporaneous relationships between the variables, we estimate the generalized impulse response functions as developed by Pesaran and Shin (1998) where the responses are order invariant.[10]

We estimate and chart the generalized impulse response functions of all three housing variables up to 10 quarters ahead with respect to one standard deviation positive shock in the funds rate and real GDP growth at time *t*. We choose a maximum of 10 periods because we are using low frequency data and more important, we are interested in the short run dynamic analysis of the responses of housing variables to funds rate or GDP growth rate over the business cycle. These graphs for the pre- and post-crisis period along with the 90% confidence interval bands, calculated based on analytical variance, are depicted below in Figures 1 and 2 respectively.

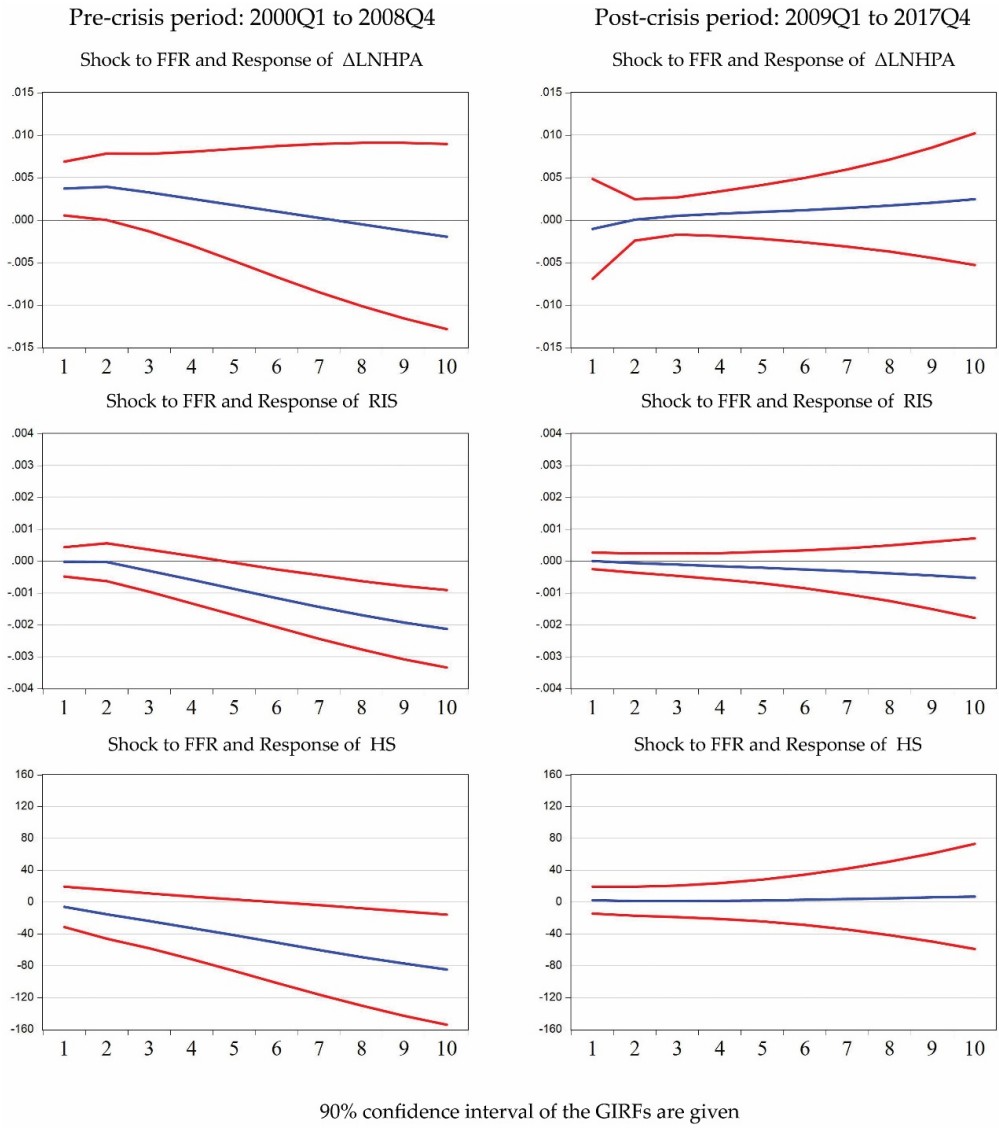

90% confidence interval of the GIRFs are given

**Figure 1.** Generalized Impulse Response Function (GIRF) of housing variables with respect to one SD positive shock to federal funds rate. FFR, ΔLNHPA, RIS and HS denote the federal funds rate, house price inflation, residential investment share in GDP and housing starts respectively.

---

[10] In orthogonalized impulse responses, the underlying shocks are orthogonalized using the Cholesky decomposition. The main criticism of Cholesky orthogonalization is that the responses change drastically if the order of the variables change as the decomposition involves a triangular matrix (see Pesaran and Shin (1998) for details).

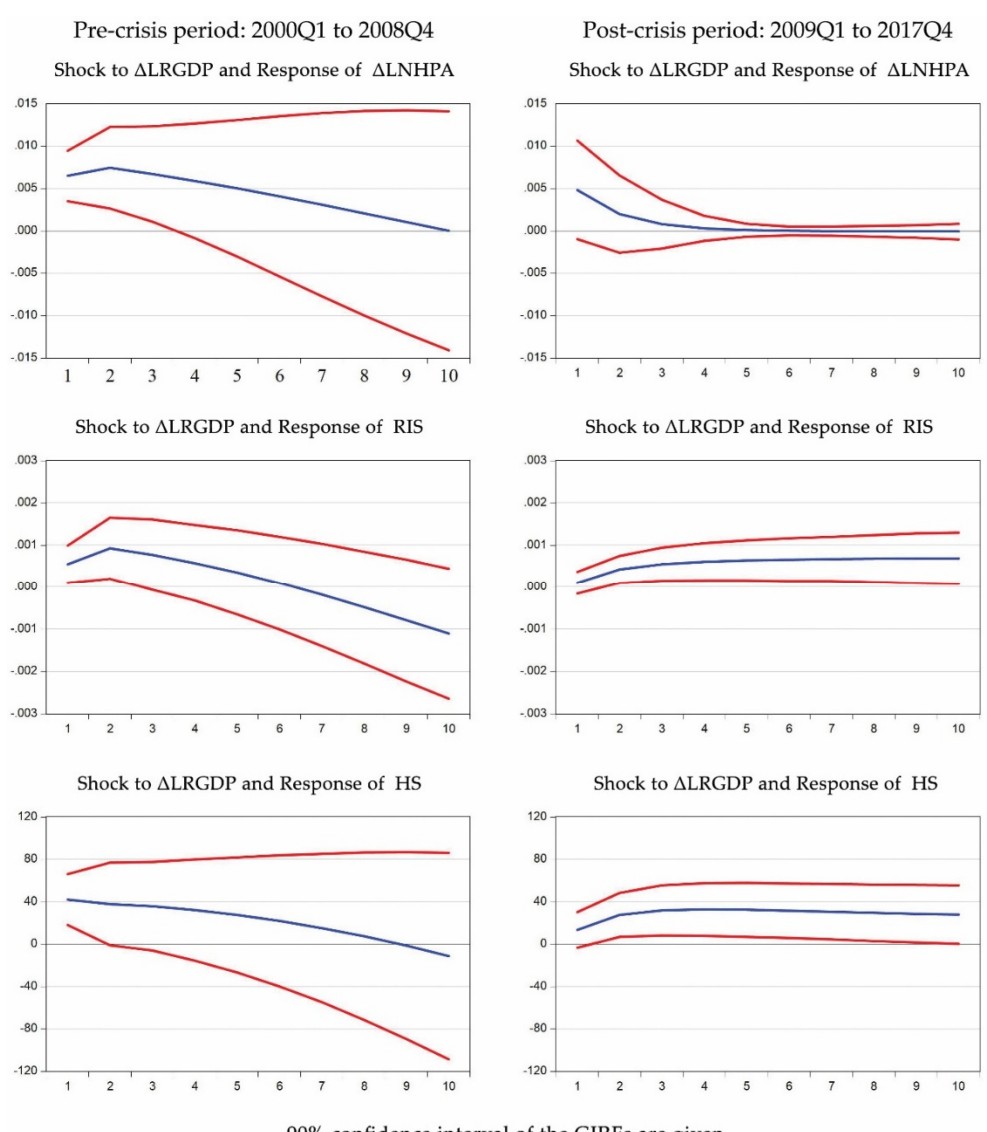

**Figure 2.** Generalized Impulse Response Function (GIRF) of housing market variable with respect to one SD positive shock to real GDP growth. ΔLRGDP, ΔLNHPA, RIS and HS denote real GDP growth, house price inflation, residential investment share in GDP and housing starts respectively.

We choose to report a 90% confidence interval band to align the impulse responses with the Granger causality analysis. As noted above based on the chi-square test statistic in Table 3, the federal funds rate causes residential investment share at 1% level of significance and the housing starts at 10% level of significance only in the pre-crisis period. However, the significance of chi-square test depends on the assumed normality of the error term. If the normality assumption does not hold, the level of significance may increase, and consequently, the 10% significance level may no longer remain statistically significant. To address this issue, we performed the Jarque and Bera normality test for all the error terms in our models and found the null hypothesis of normality rejected only in four cases. The results are given in Table A5 in the Appendix A. It is important to mention that the null hypothesis of normality of the error terms is not rejected in Model 3 which incorporates housing starts. This implies that the possibility of higher level of significance due to non-normality of error is ruled out in this case. On the other hand, while the assumption of normality is rejected in Model 2 which incorporates residential investment share, it only happens in the post crisis period when the funds rate does not cause the residential investment share to begin with.

A close inspection of Figure 1 reveals the following details. In the pre-crisis period, the house price inflation behaves differently with respect to interest rate shocks than the residential investment share or house starts. For example, a positive shock to the funds rate has a steadily growing negative effect on volume variables in the housing market with a slightly delayed response from the residential investment share. By contrast, a positive shock to the funds rate initially has a positive but diminishing impact on house price inflation which lingers up to the 7th quarter before entering the negative territory. This implies that an increase in the funds rate will decrease the volume indicators in the housing market steadily. However, the same increase in the funds rate will not reduce house prices up to 7 quarters; instead it will only decrease the rate of increase in house prices initially. This is suggestive of sluggishness in price responses in the housing market. Furthermore, the impulse response of house price inflation is statistically not significant at 10% level after the first two quarters whereas the impulse responses of residential investment share and housing starts become significant after 4 and 6 quarters respectively. In the post-crisis period, the effect of interest rate shocks on housing market variables diminishes significantly, as their impulse responses are closer to the zero line and always statistically insignificantly different from zero at 10% level. This suggests that the Federal Reserve's interest rate policy fails to explain the movements in housing market variables in the post-crisis period.[11]

As for the response of housing market with respect to one standard deviation shock in real GDP growth, it is clear from Figure 2 that in the pre-crisis period, there is a positive significant response of all three housing market variables for the first and second period ahead. The response decreases and becomes statistically insignificant afterwards with the zero-line falling within the confidence interval band. By contrast, in the post-crisis period, any positive shock in real GDP growth increases housing volume indicators steadily which is also statistically significant throughout. However, the impulse response of house price inflation is never statistically significant in the post-crisis period. Overall, our impulse response analysis reinforces our Granger causality test results demonstrating that the Federal Reserve has an impact mainly on housing volume in the pre-crisis period. Its impact weakens in the post-crisis period; instead, real GDP growth explains the housing volume.

Finally, we carry out variance decompositions to analyse fractions of forecast error variance of housing market variables that are attributable to their own past shocks versus shocks from other series in the models. The results are given in Table 4 below.

Consistent with the impulse response analysis, we carry out forecast error variance decomposition up to 10 periods ahead.[12] First, it is evident from Table 4 that the Federal Reserve has a bigger impact on housing market variables in the pre-crisis than in the post-crisis period. The proportion of forecast error variance of any housing market variable explainable by funds rate decreases from pre to post-crisis period as the forecast period continues. For example, the variance of house price inflation decreases from 7.12% to 3.64%, the variance of residential investment share decreases from 38.03% to 6.07% and the variance of housing starts decreases from 21.35% to 0.42% at 10th period ahead. Second, within the pre-crisis period, the effect of funds rate strengthens for residential investment share and house starts while it weakens for house price inflation as we move from 1st to 5th to 10th period ahead. Third, while the effect of Federal Reserve's interest policy on housing market diminishes in the post-crisis period, the contribution of real GDP growth in explaining forecast error variance of residential investment share and housing starts picks up steadily as the forecast period continues, especially at 10th period ahead. By contrast, the effect of real GDP growth in explaining variation in house price inflation decreases significantly from pre to post-crisis period. Instead, own past shocks of house prices become extremely important in the post-crisis period compared to pre-crisis period with over 90% of variance in house price inflation explained by its own past suggesting a built-in

---

[11]  Even though the federal funds rate was close to 0% during 2009–2015, if the relationship between housing variables and the funds rate prevailed in the post-crisis period, it would have shown up in the impulse responses of housing variables when a shock was given to the interest rate, after controlling for other factors.

[12]  For sake of simplicity, we are presenting results at the intervals of 1, 5 and 10 respectively.

momentum in house prices. We also computed the first order autocorrelation coefficient of house price inflation which turned out to be 0.653 with a *p* value of 0.00. This reinforces our finding of the persistent nature of house price inflation in the post-crisis period and certainly warrants further research.

**Table 4.** Forecast error Variance decomposition.

| | Variance Decomposition of ΔLNHPA | | | | | | |
|---|---|---|---|---|---|---|---|
| | **2000Q1 to 2008Q4** | | | | **2009Q1 to 2017Q4** | | |
| **Period** | **S.E.** | **FFR** | **ΔLRGDP** | **ΔLNHPA** | **S.E.** | **FFR** | **ΔLRGDP** | **ΔLNHPA** |
| 1 | 0.011811 | 9.941631 | 20.41247 | 69.64590 | 0.021438 | 0.221527 | 4.978330 | 94.80014 |
| 5 | 0.024322 | 8.397416 | 26.47954 | 65.12305 | 0.022839 | 0.540334 | 5.329699 | 94.12997 |
| 10 | 0.028097 | 7.124616 | 25.58334 | 67.29204 | 0.023209 | 3.644288 | 5.170177 | 91.18553 |
| | **Variance Decomposition of RIS** | | | | | | |
| **Period** | **S.E.** | **FFR** | **ΔLRGDP** | **RIS** | **S.E.** | **FFR** | **ΔLRGDP** | **RIS** |
| 1 | 0.001672 | 0.015145 | 14.35550 | 85.62936 | 0.000952 | 0.006853 | 1.357947 | 98.63520 |
| 5 | 0.004300 | 6.469207 | 22.97886 | 70.55193 | 0.002493 | 1.318683 | 19.84692 | 78.83440 |
| 10 | 0.006435 | 38.02760 | 12.57335 | 49.39906 | 0.003786 | 6.065616 | 23.07306 | 70.86132 |
| | **Variance Decomposition of HS** | | | | | | |
| **Period** | **S.E.** | **FFR** | **ΔLRGDP** | **HS** | **S.E.** | **FFR** | **ΔLRGDP** | **HS** |
| 1 | 92.77857 | 0.413013 | 28.71898 | 70.86800 | 61.83247 | 0.202954 | 5.018482 | 94.77856 |
| 5 | 237.0345 | 6.377980 | 22.87398 | 70.74804 | 141.9710 | 0.091849 | 20.82011 | 79.08804 |
| 10 | 359.5210 | 21.35389 | 16.52256 | 62.12355 | 192.1084 | 0.421875 | 23.92973 | 75.64839 |

FFR, ΔLNHPA, ΔLRGDP, RIS and HS denote the federal funds rate, house price inflation, real GDP growth, residential investment share in GDP and housing starts respectively.

## 5. Concluding Remarks

In this paper, we reexamine the role of the Federal Reserve in triggering the housing crisis of the past decade. We specifically focus on the stability of the relationship between monetary policy as measured by the federal funds rate and a combination of housing variables capturing both price and volume in the housing market spanning almost six decades. We add a new dimension to the existing body of literature by applying Qu and Perron (2007) structural break test for changes occurring at multiple unknown break dates in a small dimensional VAR model. We subsequently use the Qu and Perron (2007) determined endogenous break dates to split up the full sample into segments focusing on the pre- and the post crisis period for carrying out multivariate time series analysis.

The Granger causality test results indicate that the interest rate policy of the Federal Reserve did not cause house price inflation, although it did cause residential investment share and housing starts in the pre-crisis period, both being measures of volume of activity in the housing market. Our impulse response analysis reinforces these results by demonstrating sluggishness in price responsiveness in the housing market and a diminished impact of Federal Reserve policy in the post-crisis period. In addition, the impulse response graphs and the forecast error variance decomposition both point to a stronger impact of real GDP growth on housing volume in the post-crisis period while dynamics of house price inflation suggests a built-in momentum.

Overall, our findings contradict those studies that held the Federal Reserve responsible for the run-up in house prices prior to the housing market collapse. However, we do find evidence of the Federal Reserve affecting housing volume in the pre-crisis period. Our results in this respect support Leamer (2015, 2007) assertion that housing is more of a volume cycle than a price cycle. Future research can proceed by expanding our VAR framework to also include a long-term interest rate to shed light on the debate surrounding the effect of long-term vis a vis short-term rate on housing variables [see Miles (2014), McDonald and Stokes (2013c)]. Another interesting extension would be to use a more

advanced time- varying parameter VAR (TVP-VAR) model instead of a simple VAR to investigate the link between monetary policy and housing variables by considering the possible time-varying nature of the underlying structure of the macroeconomy [see He et al. (2018) and Huber and Punzi (2018) for details].

**Author Contributions:** The authors contributed equally to this paper.

**Funding:** This research received no external funding.

**Conflicts of Interest:** The authors declare no conflict of interest.

## Appendix A

**Table A1.** Results of unit root test of all the variables.

| Variable | $\Delta$LRGDP | $\Delta$LNHPA | RIS | FFR | HS |
|----------|---------------|---------------|-----|-----|-----|
| ADF | −7.26202 | −3.6379 | −2.85346 | −3.556268 | −3.36324 |
| *p*-value | (0.00) | (0.01) | (0.05) | (0.04) | (0.01) |

**Table A2.** VAR estimation results of model 1.

| | Regime 1: 2000Q1 to 2008Q4 | | | Regime 2: 2009Q1 to 2017Q4 | | |
|---|---|---|---|---|---|---|
| | $\Delta$LRGDP | $\Delta$LNHPA | FFR | $\Delta$LRGDP | $\Delta$LNHPA | FFR |
| $\Delta$LRGDP$_{t-1}$ | 0.054966 | 0.255539 | 39.55285 | 0.29813 | 0.093042 | 0.863853 |
| | (0.80) | (0.52) | (0.03) | (0.03) | (0.88) | (0.60) |
| $\Delta$LNHPA$_{t-1}$ | 0.156347 | 0.927119 | 11.72112 | 0.04125 | 0.335293 | −0.30645 |
| | (0.02) | (0.00) | (0.04) | (0.26) | (0.06) | (0.51) |
| FFR$_{t-1}$ | $-3.27 \times 10^{-5}$ | −0.00083 | 0.991962 | 0.000738 | 0.007349 | 1.19834 |
| | (0.96) | (0.45) | (0.00) | (0.81) | (0.61) | (0.00) |
| C | 0.002218 | 0.000584 | −0.51069 | 0.003188 | 0.002615 | −0.0175 |
| | (0.41) | (0.90) | (0.02) | (0.01) | (0.63) | (0.22) |

*p*-Values are given in parentheses.

**Table A3.** VAR estimation results of model 2.

| | Regime 1: 2000Q1 to 2008Q4 | | | Regime 2: 2009Q1 to 2017Q4 | | |
|---|---|---|---|---|---|---|
| | $\Delta$LRGDP | RIS | FFR | $\Delta$LRGDP | RIS | FFR |
| $\Delta$LRGDP$_{t-1}$ | 0.040554 | 0.076589 | 31.24551 | 0.37202 | 0.067732 | 0.129445 |
| | (0.84) | (0.16) | (0.03) | (0.00) | (0.01) | (0.92) |
| RIS$_{t-1}$ | 0.426163 | 1.026699 | 44.44523 | −0.013027 | 1.027946 | 6.305261 |
| | (0.00) | (0.00) | (0.00) | (0.95) | (0.00) | (0.00) |
| FFR$_{t-1}$ | −0.00013 | −0.00052 | 0.991746 | 0.001413 | −0.00072 | 1.121703 |
| | (0.82) | (0.00) | (0.00) | (0.71) | (0.36) | (0.00) |
| C | −0.01765 | −0.00058 | −2.64146 | 0.003336 | −0.00075 | −0.1932 |
| | (0.02) | (0.77) | (0.00) | (0.55) | (0.52) | (0.00) |

*p*-Values are given in parentheses.

**Table A4.** VAR estimation results of model 3.

| | Regime 1: 2000Q1 to 2008Q4 | | | Regime 2: 2009Q1 to 2017Q4 | | |
|---|---|---|---|---|---|---|
| | **ΔLRGDP** | **HS** | **FFR** | **ΔLRGDP** | **HS** | **FFR** |
| $\Delta LRGDP_{t-1}$ | −0.09751 (0.63) | −859.179 (0.79) | 24.57057 (0.13) | 0.373026 (0.00) | 3200.074 (0.05) | −0.6239 (0.65) |
| $HS_{t-1}$ | $1.41 \times 10^{-5}$ (0.00) | 1.08877 (0.00) | 0.001189 (0.00) | $-1.69 \times 10^{-7}$ (0.96) | 0.967067 (0.00) | 0.000114 (0.01) |
| $FFR_{t-1}$ | $7.16 \times 10^{-5}$ (0.89) | −13.9219 (0.10) | 1.003731 (0.00) | 0.001343 (0.70) | 4.362603 (0.93) | 1.13993 (0.00) |
| C | −0.01865 (0.00) | −124.352 (0.23) | −2.28616 (0.00) | 0.003091 (0.29) | 31.38228 (0.43) | −0.09831 (0.00) |

*p*-Values are given in parentheses.

**Table A5.** Normality test of the error term in the VAR model.

| | Pre-Crisis | | | Post-Crisis | | |
|---|---|---|---|---|---|---|
| Model 1 | | | | | | |
| | $\varepsilon_{FFR}$ | $\varepsilon_{\Delta LRGDP}$ | $\varepsilon_{\Delta LNHPAP}$ | $\varepsilon_{FFR}$ | $\varepsilon_{\Delta LRGDP}$ | $\varepsilon_{\Delta LNHPAP}$ |
| J-B test | 2.512118 | 8.217447 | 0.322027 | 5.913267 | 0.392461 | 0.205936 |
| p-value | (0.28) | (0.02) | (0.85) | (0.05) | (0.82) | (0.90) |
| Model 2 | | | | | | |
| | $\varepsilon_{FFR}$ | $\varepsilon_{\Delta LRGDP}$ | $\varepsilon_{RIS}$ | $\varepsilon_{FFR}$ | $\varepsilon_{\Delta LRGDP}$ | $\varepsilon_{RIS}$ |
| J-B test | 0.326042 | 6.850346 | 0.860979 | 3.32965 | 0.383112 | 17.01102 |
| p-value | (0.85) | (0.03) | (0.65) | (0.19) | (0.83) | (0.00) |
| Model 3 | | | | | | |
| | $\varepsilon_{FFR}$ | $\varepsilon_{\Delta LRGDP}$ | $\varepsilon_{HS}$ | $\varepsilon_{FFR}$ | $\varepsilon_{\Delta LRGDP}$ | $\varepsilon_{HS}$ |
| J-B test | 0.992582 | 1.317416 | 1.56771 | 2.666512 | 0.373458 | 0.557175 |
| p-value | (0.61) | (0.52) | (0.46) | (0.26) | (0.83) | (0.76) |

J-B test refers to Jarque Bera test for normality of the error terms.

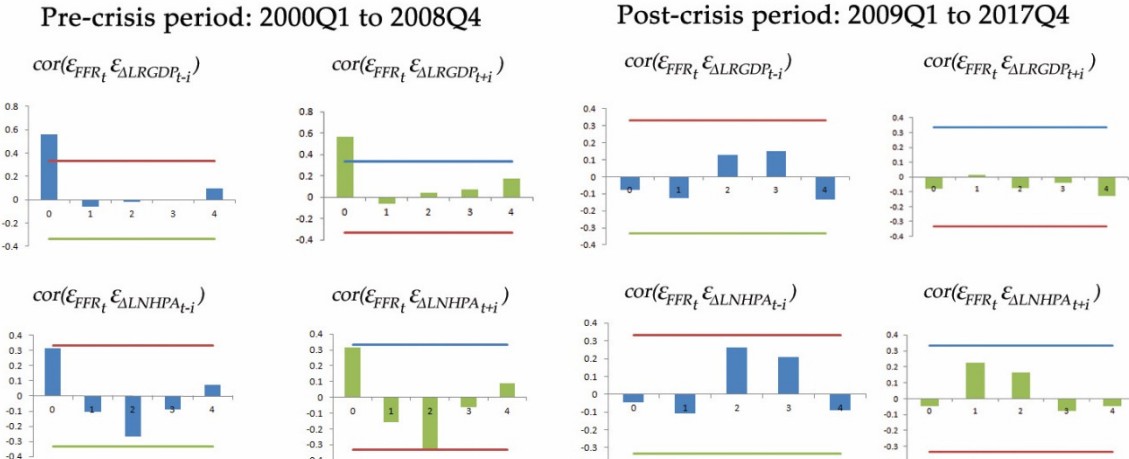

**Figure A1.** Lead and lag cross autocorrelation of the error term for Model 1.

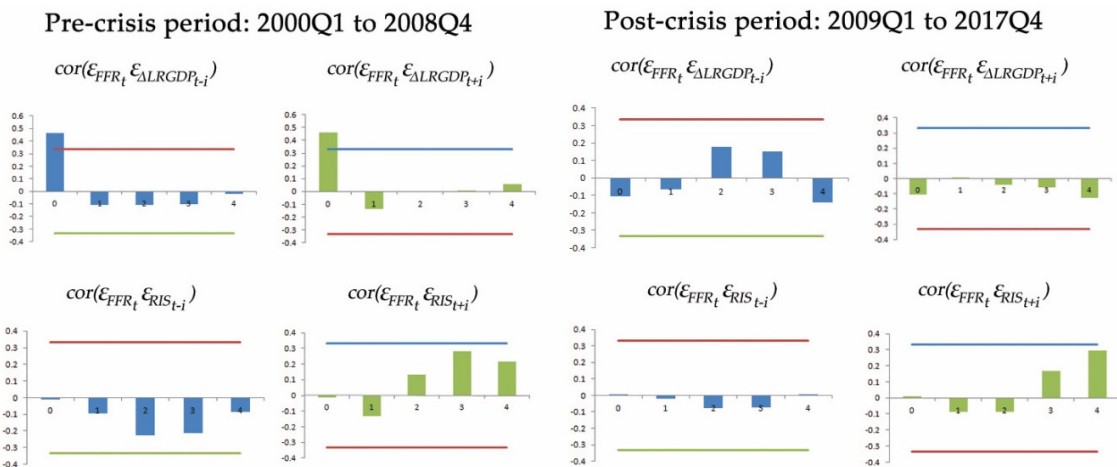

**Figure A2.** Lead and lag cross autocorrelation of the error term for Model 2.

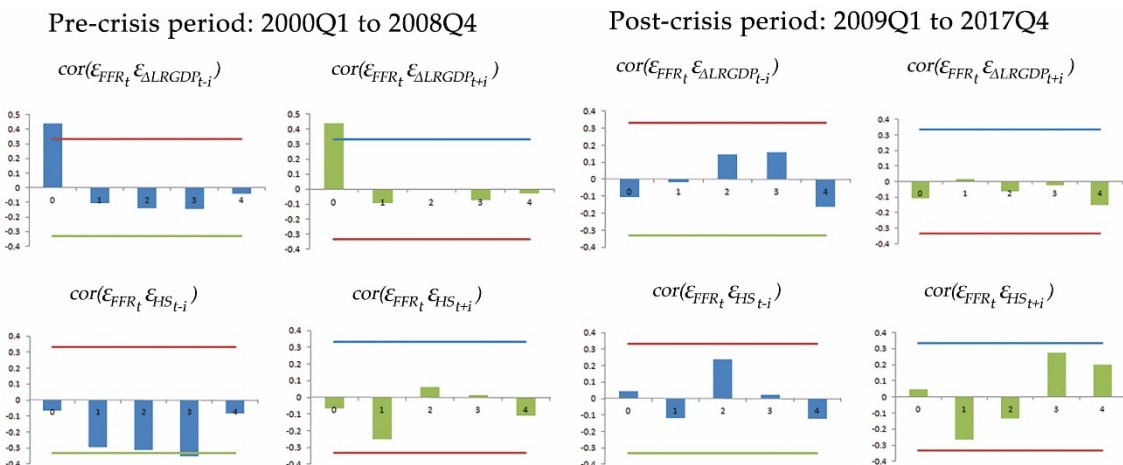

**Figure A3.** Lead and lag cross autocorrelation of the error term for Model 3.

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
