# Peer review of "The Role of the Federal Reserve in the U.S. Housing Crisis: A VAR Analysis with Endogenous Structural Breaks"

_jrfm, doi:10.3390/jrfm12030125_

Round 1
Reviewer 1 Report
In order to identify the influence of FED actions on the USA subprime housing crisis, in this paper the Authors have applied the Qu and Perron (QP) method to assess structural endogenous breaks in the sample considered, constituted by quarterly data from 1960:Q1 to 2017:Q1. The federal fund rate, the real GDP, the nominal housing prices, the share of residential investment in GDP, housing starts as an indicator of housing market activity have been the variables involved in the analysis carried out. After having identified the break periods, the Authors have implemented a classical VAR method, with the multivariate Granger causality tests, the impulse response analysis and the forecast error variance decomposition. The results have pointed out the strong relationships between the monetary policies and the housing variables over time.
The work is interesting and the methodology is well explained. The ADF tests for the verification of the stationary in the VAR models, as well as the coefficients of the equation systems that constitute the VAR models should be reported. Some considerations on the absence on "delay" analysis, i.e. on possible significant correlations among specific variable at the time t and the other ones at the time t-i, should be also developed, in order to highlight the reliability of the statistical correlations obtained.
Reviewer 2 Report
This is an interesting, ambitious attempt to address some of the biggest policy questions in the wake of the 2008-2009 global financial crisis, especially its origins in the American housing market. Although I have some misgivings about the depth and persuasiveness of the analysis, I do recommend publication.
Let's start with the article's strengths. It tells a compelling story, that of the housing bubble's contribution to the global crisis and the ongoing controversy over the Federal Reserve System's contribution to the housing bubble. As extensive as the literature on those subjects is, the authors commendably condense the background and focus on the contribution of monetary policy to the crisis.
Given the temptation of elaborating a monstrous model with dozens of potentially explanatory variables — something that happens with alarming frequency in contemporary economic literature! — the authors also deserve praise for focusing the paper's methodology on three basic tools: vector autoregression, the Qu-Perron methodology for finding structural breaks, and using Granger causality to assign responsibility (as much as social science can ever do this) to the Fed, or else to absolve the Fed of culpability for the housing crisis.
This paper does not aspire to add anything methodologically novel to the literature. And that's fine! Not every paper needs to break novel methodological ground. Indeed, the majority of the literature should consist of "normal science" — namely, investigators applying settled techniques to new data and offering novel insights. Keeping the paper under 600 lines (exclusive of crisply rendered tables, equations, and graphs), while telling a cogent story was a notable, praiseworthy achievement. As a matter of substance, confirming Edward Leamer's pre- and post-crisis pronouncement that housing cycles are indistinguishable, at least as a matter of pricing, from the broader business cycle is an important contribution. Treating housing cycles as functions of volume and not of price would advance the literature in multiple branches of economics.
Where I find myself a little uncomfortable is the strength of the story that this article tells. Model 3 in Table 3, at page 12, lines 448-450, ultimately hinges on 10% confidence, so narrowly that it comes close to rounding error for that boundary. I hardly need to remind the authors that all chi-square tests, far from being robust, are highly sensitive to violations of assumed normality. As becomes clear in the narrative exposition and the graphs on pages 13 and 14, this 90% confidence interval drives the balance of the paper's analysis.
Big claims demand big proof. The paper makes a reasonably big (albeit intuitively credible) claim that Fed policies had a greater impact on housing volume without necessarily disrupting linkages between housing prices and the broader business cycle. It's relatively rare to see authors in any branch of social science proceeding on the strength of a single finding at 90% confidence. That means that a false positive finding, based solely on normal dispersion of data, could be as high as 10%. Given the authors' admitted reliance on low-frequency data, some methodological reckoning is due. As I indicate below, I don't think that these concerns preclude publication. They do deserve some consideration and, as appropriate, some response from the authors.
There are interesting subsidiary findings throughout the paper. One of the most intriguing ones, literally thrown to the margins as though it were trivial, is the observation that housing prices follow some sort of endogenous momentum (page 16, lines 533-536). The finding of first-order autocorrelation in house price inflation at 0.653, with a p-value below 0.005 (presumably the threshold at which the authors would have rounded up to the canonical value of 0.01) deserves more than a mention in passing in footnote 27. Whether this warrants more followup in this paper, or a tantalizing promise of future research, is something I'll leave to the authors.
Finally, though it seems a bit uncharitable to suggest a closer look at the use of the English language in this paper (especially since the corresponding author holds an academic appointment in the United States), I would recommend that the authors put just a little more effort into cleaning up their language. On more than one occasion, I found myself retracing a sentence just to make sure that it said what I thought the authors intended. In multiple instances, the authors made poor use of "the," "an," and "a" — which, in their defense, are often as subtle as they are common. Let's start with the title. Please: The Role of the Federal Reserve in the U.S. Housing Crisis …. The omission of "the" from titles, so that they sound "title-like" rather than short, understandable snippets of ordinary language, just sounds illiterate. So do sentences such as these:
"we carry out multivariate time series analysis for pre and post-crisis period" (abstract, line 16) — Should be: "we carry out multivariate time analysis for pre- and post-crisis periods."
"Our study … focuses on loose monetary policy of the Fed. … McDonald and Stokes (2013b) provide an extensive literature review for interested reader." (lines 42-45) — Should be: "the loose monetary policy of the Fed" or "the Fed's loose monetary policy." Should be: "an extensive literature review for the interested reader" or "for interested readers."
Yes, this is trivial stuff. It also drove me to distraction as I worked through the paper.
I also take exception to the use of the phrase "Granger cause" as though it were a verb. It emphatically is not. In the abstract, you simply must render this sentence properly. I suggest: "Our interpretation of Granger causality indicates that the Fed did not cause house price inflation. By the same token, Granger causality analysis suggests that the Fed was responsible for raising residential investment share …" And so on. The fact that scientific journals now use English to reach a global audience raises rather than lowers the premium on proper use of the language.
The authors are more than competent linguistically. Instead of marking them down and asking for moderate changes (which strikes me as a far harsher judgment than I intend to convey), I am content to exhort them to pay a little more attention to what may be the easiest detail to resolve before publication.
Reviewer 3 Report
General comments
It is very difficult to understand the objectives of the paper. Structural break, causality tests, VAR are used. The authors have to highlight the relationship between them and why these are adopted when testing structural break can be done by multiple regression as well.
The authors have to polish English and some of the footnotes can be put back to the main paper. It is not necessary to have so many footnotes which increase the readers’ time in reading the paper back and forth.
Specific comments
Abstract, what is the implication of "We find evidence of several structural breaks with the most recent one occurring during the 4 15 th quarter of 2008"
Abstract, what is the volume cycle? Why do we need to know the structural break? What is the importance of the structural break? What are the policy and practical contribution?
Abstract, what aspect of Fed do you refer to? Policy? Interest rate? Please state.
Introduction, please add the citations for Several researchers. Who are they?
Footnote 1, wrong format. Please fix that.
Line 42, what is the relationship of "Our study is based on the explanation that focuses on loose monetary policy of the Fed." with structural break?
Line 50, what is accommodative monetary policy?
Line 52-53, what does it mean by "held to the middle ground"?
Line 68, what was the major finding of "Miles (2014) performed the Andrews-Quandt endogenous structural break test (1993)"?
Line 78, while VAR is well-known to everybody, what is parsimonius vector autoregression?
Please state and explain. Delete "setup".
Line 85, QP (2007), wrong citation?
Line 97, Model 2 and 3, should be Models...
Footnotes 3-4, wrong format, please check all the footnote formats according to MDPI format.
Line 144, please put the footnote back to the main document.
Line 169, effect of transferring building backward? What does it mean?
Line 233-234, “We concur with Miles that those break points in the relationship need to be determined 234 endogenously within the model” Why?
Section 2, please state the hypotheses, objective and gap of the research that this paper aims to fill in view of the previous literature.
Lines 262-274, please check whether QP is appropriate citation.
Lines 341-346, why nominal but not real? Please state the reasons.
Lines 368-369, why The residential investment share is calculated as the ratio of nominal residential investment to nominal GDP? Again, why nominal? Nominal includes inflation problem.
Footnotes, 9-11 wrong format of citations and references.
Footnotes 21-22, no citations. Please add.
Line 546, “Fed did not Granger cause house price inflation” the fed in itself cannot cause house inflation of cause, the authors have to state what aspect of the fed…
Suggested references to be included:
Bank Credit and Housing Prices in China: Evidence from a TVP-VAR Model with Stochastic Volatility https://www.mdpi.com/1911-8074/11/4/90
https://www.mdpi.com/2071-1050/10/2/341/htm (real estate rate and various factors that affect housing prices)
The preliminary effectiveness of bilateral trade in China’s belt and road initiatives: a structural break approach https://www.tandfonline.com/doi/full/10.1080/00036846.2019.1584387
Round 2
Reviewer 3 Report
General comments
It seems the authors have not carefully respond to all the comments given by the reviewer. I would strongly suggest the authors check ALL the comments made by the reviewers and revise the paper accordingly but not selectively respond to some of the comments which the authors try to respond. There are serious grammatical errors which made the whole paper very difficult to read and follow. Make sure that the author understand that methodology and method are two different terms.
Specific comments
The abstract still has a lot of flaws in terms of academic writing. For example, "Specifically, we ask if the relationship between the federal funds rate and the housing 12 variables underwent structural changes in the wake of the housing crisis" The word "ask looks strange in abstract.
Line 17, endogenize structural changes, how to "endogenize" the structural change?
Line 18, methodology or method? Please note that these two words have different meaning.
Line 22-24, Overall, our findings support Leamer (2015, 2007) that housing volume fluctuates more than house prices over the business cycle...if your finding simply support the previous research, why do we need your "new" paper? What is the point of innovation? Originality?
Lines 58-59, "Our study is based on the fourth explanation that focuses on the loose monetary policy of the Fed." What is the fourth explanation?
Lines 62-63, "ask if the relationship between the federal funds rate, real GDP and the housing variables has been stable for all time periods or if there have been structural breaks in the relationship over time." Why do we need to ask? The term is also inappropriate here.
Line 71, defended the low federal funds rate? How to defend?
Footnotes 1-13, why 13 footnotes are needed? Why not put that back to the main paper body?
Line 91, if "He found evidence of structural breaks in the impact of federal funds rate", what will be the contribution of this paper?
Line 99, wrong citation format; methodology? Or method?
Line 159, Federal Reserve instead of Fed.
Line 161, polish the sentence "both being measures of volume" .
Line 198, excessive and unnecessary too much references, McDonald and Stokes (2013a,2013b,2013c,2015) from the same authors and not sure what are the value added of these citations when they are packed together.
Line 289, and that too?
Try to avoid too many abbreviations terms like QP. Authors do not want to flip back to check what it is.
Lines 292-294, "Ahamada and Sanchez (2013) applied the QP test to study the U.S. house price – macro link over time using a VAR framework. Using quarterly data from 1960:Q1 to 2009:Q3, they found evidence of two break points in the house price" This indicate there was similar research as before, why do we need your paper? Same problem observe in line 503.
Lines 363-386, please revise the font size of the math formula.
Again, line 428, not sure why (2013a,2013b,2013c,2015) are cited?
Line 482, wrong font size
Line 494, please state, what happened in1969:Q1,1987:Q1, 2000:Q1 and 2008:Q4?
Lin 496, what happens to1969:Q1, 1977:Q4, 1987:Q1, 1999:Q4 and 2008:Q4? While mathematically we may see many breaks, we also want to know why so has to predict the future.
Line 503, what is your contribution that makes different from that paper in 2013? Apart from the date is different?
Line 605, wrong format.
Figure 1, wrong font size.
Please check all the table format according to MDPI.
Lines 672-673, "The Granger causality test results indicate that the interest rate policy of the Fed did not Granger cause house price inflation" Why?
References, please check all the styles, some are using those not in MDPI format. Please reread the reviewer's suggestion on references to be added:
https://www.mdpi.com/2071-1050/10/2/341/htm (real estate rate and various factors that affect housing prices)
The preliminary effectiveness of bilateral trade in China’s belt and road initiatives: a structural break approach https://www.tandfonline.com/doi/full/10.1080/00036846.2019.1584387
And add more references from 2018/2019.
Round 3
Reviewer 3 Report
The authors have not fully responded to the reviwers' suggestions.
Line 58-60, highlight the paper lack of originality loose monetary policy of the Federal Reserve. We examine whether the Fed’s interest rate policy of Federal Reserve played any role in the formation of the housing bubble.
It is pretty well-known that Central bank's ill policy lead to bubble.
The research method in itself cannot show any novel or innovative idea as well.
Many of these lack of literature support, e.g. There is a growing body of literature in line 43 fail to shows what are these literature in the whole paragraph. It seems to us that this claim is made by the author only. Who made this claim is a question.
Line 15-18, While previous studies mostly set break-dates based on events known a priori to split the full sample to subsamples, we endogenouslyize determine structural break points changes occurring at multiple unknown break 18 dates by applying Qu-Perron (2007) methodology...as I mentioned in my review many times, why do we need to know these breaks?
Line 19, the federal funds rate did not cause house price inflation, why?
Line 21-22, while house price inflation had a momentum of its own? What does it mean?
The whole paper has not been copy edited despite I said that many times that the paper needs copy edit.
All in all, the authors go on their own way and fail to respond to the reviewers' comments. It is very difficult to read and cannot show why their paper has to be published. I think it is well-deserved to put that to reject.